# PROACTIVE SEQUENCE GENERATOR VIA KNOWLEDGE ACQUISITION

## ABSTRACT

Sequence-to-sequence models such as transformers, which are now being used in a wide variety of NLP tasks, typically need to have very high capacity in order to perform well. Unfortunately, in production, memory size and inference speed are all strictly constrained. To address this problem, Knowledge Distillation (KD), a technique to train small models to mimic larger pre-trained models, has drawn lots of attention. The KD approach basically attempts to maximize recall, *i.e.*, ranking Top-$k$" tokens in teacher models as higher as possible, however, whereas precision is more important for sequence generation because of exposure bias. Motivated by this, we develop **K**nowledge **A**cquisition (KA) where student models receive $\log \mathbf{q}(\mathbf{y}_t|\mathbf{y}_{<t}, \mathbf{x})$ as rewards when producing the next token $\mathbf{y}_t$ given previous tokens $\mathbf{y}_{<t}$ and the source sentence $\mathbf{x}$. We demonstrate the effectiveness of our approach on WMT'17 De-En and IWSLT'15 Th-En translation tasks, with experimental results showing that our approach gains +0.7-1.1 BLEU score compared to token-level knowledge distillation.

## 1 INTRODUCTION

We have recently witnessed rapid progress sequence models for natural language processing (NLP). In particular, transformers (Vaswani et al., 2017; Dai et al., 2019) and BERT (Devlin et al., 2018; Liu et al., 2019) have been established as state-of-the-art approaches for sequence generation and language modeling respectively. However, the resulting sequence models are often exceedingly large. For example, a big transformer with 6 layers, 1024 dim and 16 heads has $\sim$ 100M parameters, which can make production deployment impractical, especially on mobile devices.

Unfortunately, learning small sequence models is very challenging because of the nature of deep learning - curve fitting. Training deep neural networks is data hungry and needs bigger "brains". To deal with the challenges, Kim & Rush (2016) consider applying Knowledge Distillation approaches (Hinton et al., 2015) to match the distribution of the teacher model. At token-level, KD approaches minimize $\mathbb{D}_{KL}(\mathbf{q}(\mathbf{y}_t|\mathbf{y}_{<t}, \mathbf{x})\|\mathbf{p}(\mathbf{y}_t|\mathbf{y}_{<t}, \mathbf{x}))$ where $\mathbf{q}(\mathbf{y}_t|\mathbf{y}_{<t}, \mathbf{x})$ [teacher] or $\mathbf{p}(\mathbf{y}_t|\mathbf{y}_{<t}, \mathbf{x})$ [student] is a probability distribution over tokens in the vocabulary conditioned on previous tokens $\mathbf{y}_{<t}$ and the source sentence $\mathbf{x}$. In other words, the KD approaches maximize **recall** - the fraction of the top-$k$" tokens in $\mathbf{q}(\mathbf{y}_t|\mathbf{y}_{<t}, \mathbf{x})$ have been retrieved among the top-$k'$ in $\mathbf{p}(\mathbf{y}_t|\mathbf{y}_{<t}, \mathbf{x})$. However, sequence modeling actually cares about **precision** - the fraction of the top-$k'$ tokens in $\mathbf{p}(\mathbf{y}_t|\mathbf{y}_{<t}, \mathbf{x})$ are included in the top-$k$" of $\mathbf{q}(\mathbf{y}_t|\mathbf{y}_{<t}, \mathbf{x})$, because imperfect predictions in the top-$k'$ cause exposure bias at inference time.

To optimize precision, we attempt to minimize KL-divergence in the reverse order, that is, $\mathbb{D}_{KL}(\mathbf{p}(\mathbf{y}_t|\mathbf{y}_{<t}, \mathbf{x})\|\mathbf{q}(\mathbf{y}_t|\mathbf{y}_{<t}, \mathbf{x}))$. Our approach can be thought of as a kind of Actor-Critic approach where the "Actor" is a student model asking for advice when learning to generate every single token and the "Critic" is a teacher model that gives $\log \mathbf{q}(\mathbf{y}_t|\mathbf{y}_{<t}, \mathbf{x}))$ as rewards. After receiving the feedback, "Actor" updates it's belief accordingly, as shown in Fig. 1.

Unlike task-specific metrics such as BLEU score, teacher models summarize training data points and structure knowledge in an abstract, hierarchical manner. Therefore, they are able to offer more useful advice than matching n-grams with a finite number of human references. For example, assume there is a training example {SRC: "Die Landschaft am Meer ist wunderschn", TRG: "Amazing view along the sea" }. During learning, given the first token "Amazing", student models might ask "How about saying amazing view?", teacher models most likely give high rewards since it's expected to go with

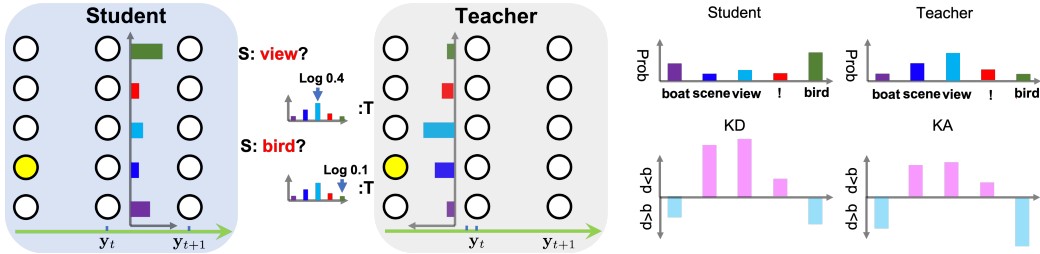

(a) Token-level Matching.                 (b) Learning strategy.

Figure 1: Student-Teacher learning system. (a) Given previous tokens (yellow), the student model asks feedback for producing a token, *e.g.*, "view", and the teacher model gives rewards, *e.g.*, $\log \mathbf{q}$("view"|"Amazing", $\mathbf{x}$). (b) The difference in learning strategy matters! We ignore $\mathbf{y}_{<t}$ and $\mathbf{x}$ for simplicity. We flip the sign of gradients and demonstrate them in the 2nd row. When $\mathbf{q}(\mathbf{y}_t)$ dominates $\mathbf{p}(\mathbf{y}_t)$, the gradients less than 0 (magenta) and $\mathbf{p}(\mathbf{y}_t) \uparrow$. In contrast, when $\mathbf{p}(\mathbf{y}_t)$ dominates $\mathbf{q}(\mathbf{y}_t)$, the gradients greater than 0 (cyan) and $\mathbf{p}(\mathbf{y}_t) \downarrow$. More importantly, KA pulls the probability of "view" and "scene" more gently but push the probability of "bird" and "boat" much harder compared to KD. In other words, KD attempts to rank "view", "scene" higher to increase recall, while KA aims at reducing the chance of "bird" and "boat" being picked to maximize precision.

"Amazing view

$$
\begin{pmatrix}
\texttt{of the seaside,} & \texttt{of the boats in the sea} \\
\texttt{of the beach,} & \texttt{from the boats} \\
\texttt{...,} & \texttt{...} \\
\texttt{along the sea,} & \texttt{of the birds flying over the water}
\end{pmatrix}
$$

", where most of them express similar meanings with the human references even though they have mismatched n-grams.

**Contributions.** To the best of our knowledge, we are the first to provide a depth-analysis of the effects of minimizing $\mathbb{D}_{KL}(\mathbf{p}\|\mathbf{q})$ on alleviating exposure bias. In addition, we enable stable reinforcement training via replacing task-specific metrics with high-capacity models. Finally, we improved +1.1 BLEU score on WMT'17 de-en task and +0.7 BLEU score on IWSLT'15 th-en task over token-level KD with up to relative 200% improvement.

## 2    SEQUENCE GENERATION

Let $(\mathbf{x}, \mathbf{y}) \in (\mathcal{X}, \mathcal{Y})$ be an example translation pair, where $\mathbf{x}$ is the source sentence and $\mathbf{y}$ is the target sentence. $\mathbf{y}_t$ is the token at position $t$ and $\mathbf{y}_{<t}$ is tokens before position $t$. Sutskever et al. (2014) propose sequence-to-sequence models which typically factorize the joint probability $\mathbf{p}(\mathbf{y} \mid \mathbf{x})$ over a sequence of conditional probabilities with parameter $\theta$:

$$
\mathbf{p}(\mathbf{y}|\mathbf{x}) = \prod_{t=0}^{T} \mathbf{p}_\theta(\mathbf{y}_t \mid \mathbf{y}_{<t}, \mathbf{x}) \tag{1}
$$

where, $\mathbf{p}_\theta(\mathbf{y}_t \mid \mathbf{y}_{<t}, \mathbf{x})$ is the probability of the token $\mathbf{y}_t$ given previous tokens $\mathbf{y}_{<t}$ and the source sentence $\mathbf{x}$. Basically, the preceding tokens $\mathbf{y}_{<t}$ are encoded into the hidden states via a state transition function

$$
\mathbf{h}_t = f(\mathbf{h}_{t-1}, \mathbf{y}_{t-1}; \mathbf{x}) \tag{2}
$$

By substituting Eqn.(2) for Eqn.(1), we have

$$
\mathbf{p}(\mathbf{y}|\mathbf{x}) = \prod_{t=0}^{T} \underbrace{\mathbf{p}_\theta(\mathbf{y}_t \mid \mathbf{h}_t)}_{\text{Policy } \pi} \tag{3}
$$

This tells us that the sequence models, e.g., RNNs, acts like a stochastic policy which picks a discrete action, *i.e.*, producing a token $\mathbf{y}_t \in \mathcal{V}$, running on a world model $\mathcal{M}$ with transition function $f$.

**Training.** We minimize the cross-entropy loss

$$
\mathcal{L}_{CE} = -\sum_t \log \mathbf{p}_\theta(\mathbf{y}_t^{ref} \mid \mathbf{y}_{<t}^{ref}, \mathbf{x}) \tag{4}
$$

where, $\mathbf{y}^{ref}$ denotes human references. At each position, the models are conditioned on the ground-truth tokens annotated by humans no matter what tokens are predicted by themselves.

**Inference.** In sequence generation tasks, exact inference is intractable due to exponentially large search space. Instead, we apply an approximation inference algorithm - Beam Search (BS). BS is a greedy heuristic search that maintains the top-B most likely partial sequences through the search tree, where B is referred to as the beam size. At each position, BS expands these B partial sequences to all possible beam extensions and then selects the B highest scoring among the expansions. Unlike training, ground-truth tokens are not available. The models have to use their own predictions in decoding.

**Evaluation.** To evaluate the quality of generated sequences, we typically use metrics such as BLEU score to measure their n-grams overlap with human references. However, the standard metrics are problematic and none of them correlate strongly with human evaluation at the level of individual sentences. For example, given a human reference "Amazing view along the sea", the sequence "The scenery of the seaside is beautiful" gets low BLEU score because there is no matching n-grams of order 2, 3, or 4. In addition, human references are a few sentences lacking in diversity. For example, when asking more people, they might say "A nice beach" or " what amazing view of the seashore".

## 2.1 DISCREPANCY AMONG PROCEDURES

**Exposure bias.** During training, the models only explore the training data distribution, but never get exposed to their own predictions. Searching in under-explored space causes errors. More importantly, such errors accumulate over time because of greedy search. For the example in Sec. 1, assume there are no sentences in training data starting with "Amazing".

> At position 1: pick a token from $\mathbf{p}(\mathbf{y}|\text{"Amazing"}, \mathbf{x}) \Rightarrow$ "cup"
>
> At position 2: pick a token from $\mathbf{p}(\mathbf{y}|\text{"Amazing cup"}, \mathbf{x}) \Rightarrow$ "on"
>
> At position 3: pick a token from $\mathbf{p}(\mathbf{y}|\text{"Amazing cup on"}, \mathbf{x}) \Rightarrow$ "table"

We can see that the poor token "cup" caused by the noisy distribution $\mathbf{p}(\mathbf{y}|\text{"Amazing"}, \mathbf{x})$ makes the situation even worse. The distributions become more and more noisy and the generation goes far away.

**Sub-optimal models.** The training objective is different with the metrics used in evaluation. To address this issue, some works attempt to directly optimize the metrics. It definitely helps improving the score, but we suspect that this might hurt the models because poor metrics discourage learning the semantic meanings. For example, the low score of "The scenery of the seaside is beautiful" makes the meanings of "Amazing" and "beautiful" or "seaside" and "sea" far from the other.

## 3 KNOWLEDGE ACQUISITION

Unlike knowledge distillation, we attempt to minimize

$$\mathbb{D}_{KL}\Big[\mathbf{p}_\theta(\mathbf{y}_t \mid \mathbf{h}_t)\|\mathbf{q}_\phi(\mathbf{y}_t \mid \mathbf{h}_t)\Big] \tag{5}$$

$$= -\Big[\mathbb{E}_{\mathbf{y}_t \sim \mathbf{p}_\theta(\mathbf{y}_t|\mathbf{b}_t)}\big[\log \mathbf{q}_\phi(\mathbf{y}_t \mid \mathbf{h}_t)\big] + \mathbb{H}\big[\mathbf{p}_\theta(\mathbf{y}_t \mid \mathbf{h}_t)\big]\Big] \tag{6}$$

Eqn.6 can be explained as an Actor-Critic algorithm (Konda & Tsitsiklis, 2000) that maximizes the expected rewards with an entropy term, where

> **"Critic":** (aka teacher models) estimate the rewards of producing token $\mathbf{y}_t$ given preceding tokens $\mathbf{y}_{<t}$ and the source sentence $\mathbf{x}$, *i.e.*,
>
> $$Q(a = \mathbf{y}_t, s = \mathbf{h}_t) = \log \mathbf{q}_\phi(\mathbf{y}_t \mid \mathbf{h}_t) \tag{7}$$
>
> **"Actor":** updates the policy $\mathbf{p}_\theta(\mathbf{y}_t \mid \mathbf{h}_t)$ guided by the "Critic"

As opposed to behavior cloning, student models ask teacher models to evaluate their behaviors and then update accordingly when producing the next token. By comparing with KD that attempts to minimize $\mathbb{D}_{KL}\big[\mathbf{q}\|\mathbf{p}\big]$, our approach is actually a kind of `reverse` KD, which we call "Knowledge Acquisition" (KA).

## 3.1 LEARNING STRATEGY

Without loss of generality, we simplify the problem to a 1-D problem where the student model $\mathbf{p}(x)$ is a single-modal Gaussian and the teacher model $\mathbf{q}(x)$ is a mixture of two Gaussians, as shown in

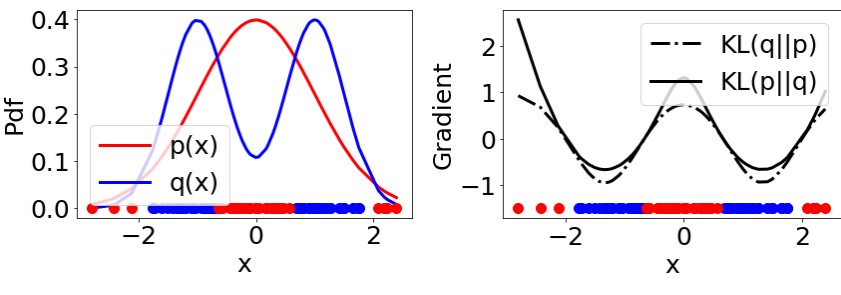

(a) $\mathbf{p}(x)$ and $\mathbf{q}(x)$        (b) Gradients

Figure 2: Matching two 1-D distributions. (a) $\mathbf{p}(x)$ and $\mathbf{q}(x)$. (b) The derivatives of $\mathbb{D}_{KL}$ w.r.t $\mathbf{p}(x)$ in both orders. $\mathbb{D}_{KL}(\mathbf{q}(x)\|\mathbf{p}(x))$ pulls more on $x$ where $\mathbf{q}(x) > \mathbf{p}(x)$ (blue region) while $\mathbb{D}_{KL}(\mathbf{p}(x)\|\mathbf{q}(x))$ pushes more on $x$ where $\mathbf{p}(x) > \mathbf{q}(x)$ (red region).

Fig. 2. Let's take a depth-analysis of their objective functions and their derivatives w.r.t. $\mathbf{p}(x)$ to see what changes after swapping $\mathbf{p}(x)$ and $\mathbf{q}(x)$.

### 3.1.1 OBJECTIVE FUNCTIONS

The entropy term in Eqn.(6) goes away in KD because $\mathbf{q}(x)$ is a fixed value and never be optimized during training. It's clear that the entropy term helps to avoid over-fitting the model to human references and generalize well on unseen test data.

### 3.1.2 DERIVATIVES

Based on Lagrangian relaxation (see Appendix A.1), the derivatives of $\mathbb{D}_{KL}$ w.r.t. $\mathbf{p}(x)$ in both orders are

$$\frac{\partial \mathbb{D}_{KL}(\mathbf{q}(x)\|\mathbf{p}(x))}{\partial \mathbf{p}(\mathbf{x})} = \underbrace{1 - \frac{\mathbf{q}(x)}{\mathbf{p}(x)}}_{G_{\mathbf{q}\|\mathbf{p}}(x)}, \quad \frac{\partial \mathbb{D}_{KL}(\mathbf{p}(x)\|\mathbf{q}(x))}{\partial \mathbf{p}(\mathbf{x})} = \underbrace{\log \frac{\mathbf{p}(x)}{\mathbf{q}(x)}}_{G_{\mathbf{p}\|\mathbf{q}}(x)} \tag{8}$$

Eqn.(8) tells us that

(a) $\mathbf{q}(x) > \mathbf{p}(x) \Rightarrow G_{\mathbf{q}\|\mathbf{p}}(x) < 0$ and $G_{\mathbf{p}\|\mathbf{q}}(x) < 0$

(b) $\mathbf{q}(x) < \mathbf{p}(x) \Rightarrow G_{\mathbf{q}\|\mathbf{p}}(x) > 0$ and $G_{\mathbf{p}\|\mathbf{q}}(x) > 0$

That is, the probability of $\mathbf{x}$ is pulled up when $\mathbf{q}(x)$ dominates $\mathbf{p}(x)$, and pushed down otherwise. However, they take efforts at different levels (see Appendix A.2).

In (a), $\left|G_{\mathbf{q}\|\mathbf{p}}(x)\right| > \left|G_{\mathbf{p}\|\mathbf{q}}(x)\right|$ while in (b), $\left|G_{\mathbf{q}\|\mathbf{p}}(x)\right| < \left|G_{\mathbf{p}\|\mathbf{q}}(x)\right|$

In other words, minimizing $\mathbb{D}_{KL}(\mathbf{q}(x)\|\mathbf{p}(x))$ attempts to pull up the probability of $\mathbf{x}$ that underestimated by the student models while minimizing $\mathbb{D}_{KL}(\mathbf{p}(x)\|\mathbf{q}(x))$ attempts to push down the the probability of $\mathbf{x}$ that was over-estimated by the student models. If we define

$$\text{Recall}_{k'}(\mathbf{p}(x), k") = \qquad\qquad \text{Precision}_{k"}(\mathbf{p}(x), k') =$$
$$\frac{|\mathbf{p}'s\ \text{Top-}k' \cap \mathbf{q}'s\ \text{Top-}k"|}{k"} \tag{9} \qquad \frac{|\mathbf{p}'s\ \text{Top-}k' \cap \mathbf{q}'s\ \text{Top-}k"|}{k'} \tag{10}$$

where, $\mathbf{p}'s$ Top-$k'$ is a set including $k'$ items having the highest rankings in $\mathbf{p}(x)$ and $\mathbf{q}'s$ Top-$k"$ is a set including $k"$ items having the highest rankings in $\mathbf{q}(x)$. Clearly, minimizing $\mathbb{D}_{KL}(\mathbf{q}(x)\|\mathbf{p}(x))$ is to maximize **recall** because its role is to increase the chance that $\mathbf{x}$'s in $\mathbf{q}'s$ Top-$k"$ are truly picked. By contrast, minimizing $\mathbb{D}_{KL}(\mathbf{p}(x)\|\mathbf{q}(x))$ is to maximize **precision** since its role is to avoid $\mathbf{x}$'s that are over-estimated being picked by mistake.

Overall, sequence modelling cares about **precision** because at test time, the generation is sensitive to the model's predictions. The errors caused by inaccurate predictions in the top-$k'$ accumulate and make the generation brittle.

### 3.1.3 DOES THE DISTINCTION MATTER?

One might reasonably ask whether the distinction matters, given that both learning strategies aim to eventually reach the same equilibrium point, i.e., $\mathbf{p}(x) = \mathbf{q}(x)$? In practice, the answer is yes due to the following reasons:

**Low-capacity $\mathbf{p}(x)$.** In practice, student models are of limited capacity due to memory and inference speed restrictions, especially in production. For example, running a big transformer on mobile device with real-time response is impractical. In this case, the equilibrium state $\mathbf{p}(x) = \mathbf{q}(x)$ is unreachable. Thus, different learning strategies lead to different local optima, and result in different performance.

**Imperfect $\mathbf{q}(x)$.** The teacher models are generally better approximations of the real underlying distributions from which the training data is, but they are still not 100% accurate. In particular, the exponentially-large search space makes sequence models easily over-fit to more frequently occurring data points, while being uncertain about the data points that are less likely observed. Kim & Rush (2016) also point out a mixture of $\mathbb{D}_{KL}(\mathbf{q}\|\mathbf{p})$ and cross-entropy loss reaches the best performance.

In summary, with the limited capacity and noisy $\mathbf{q}(x)$, picking what to learn smartly turns out to be important.

## 3.2 TO BE PROACTIVE WITH RICH ADVICE

To further study the capability in exploration, we *explicitly* restrict the knowledge to be partially observed by allowing only the top-$k$ to be available. The $\mathbb{D}_{KL}$ term becomes

$$\sum_{\mathbf{x} \in \text{Top-}k\ \mathbf{p}(\mathbf{x})} \mathbf{p}(\mathbf{x})\big(\log \mathbf{p}(\mathbf{x}) - \log \mathbf{q}(\mathbf{x})\big)$$

where, Top-$k$ $\mathbf{p}(\mathbf{x})$ is a set of $k$ items with the highest ranking. Only $\mathbf{p}(\mathbf{x})$ and $\mathbf{q}(\mathbf{x})$ according to the $\mathbf{x}$ in the top-$k$ are calculated while the rest is discarded. Assume $\mathbf{x}$ is a discrete variable and takes value 0, 1, 2, and 3. We build a quite simple model with a single soft-max layer to produce a distribution over $\mathbf{x}$, and set top-$k$=2. The real distribution is

$$\mathbf{q}_0 = 0.4,\ \mathbf{q}_1 = 0.3,\ \mathbf{q}_2 = 0.2,\ \mathbf{q}_3 = 0.1$$

Fig. 3 shows that KD drives $\mathbf{p}_0$ and $\mathbf{p}_1$ to the real values very quickly, but almost ignore $\mathbf{p}_2$ and $\mathbf{p}_3$ since they are not directly optimized. In contrast, KA first optimizes $\mathbf{p}_0$ and $\mathbf{p}_2$ which are the top-2. After $\sim 100$ epochs, when $\mathbf{p}_0$ and $\mathbf{p}_1$ become the top-2, the loss drops very fast and even lower than that of KD. Moreover, KA drives $\mathbf{p}_2$ and $\mathbf{p}_3$ much closer to the real values since the model has explored 3 states, i.e., $\mathbf{x} = 0, 1, 2$.

In sequence generation tasks, if student models receive rewards at each position, the entire distribution is fully visible. KA implicitly explores in the above way.

In theory, we can generalize our approach to multi-step

$$\mathbb{E}_{\mathbf{y}_{t:t'} \sim \mathbf{p}_\theta(\mathbf{y}_{t:t'}|\mathbf{h}_t)}\big[\log \mathbf{q}_\phi(\mathbf{y}_{t:t'} \mid \mathbf{h}_t)\big] + \mathbb{H}\big[\mathbf{p}_\theta(\mathbf{y}_{t:t'} \mid \mathbf{h}_t)\big] \tag{11}$$

i.e., minimizing $\mathbb{D}_{KL}\big(\mathbf{p}_\theta(\mathbf{y}_{t:t'}|\mathbf{h}_t)\|\mathbf{q}_\phi(\mathbf{y}_{t:t'}|\mathbf{h}_t)\big)$. $\mathbf{y}_{t:t'}$ are tokens starting with position $t$ and ending at position $t'$. By increasing $t'$-$t$, the student models can capture long-term dependency and are much more capable of handling exposure bias. However, the training is expensive even with $t'$-$t$=2 since the search space explodes exponentially. In this case, picking top-$k$ candidates is a solution. However, in practice we only model the reward for a single step (but use the signal from all steps in each batch update).

## 4 RELATED WORK

Our work closely relates to two lines of work: knowledge transfer and dealing with the mismatch between training and inference procedures. Both of them have already been explored in the literature.

**Learning $\neq$ inference.** To handle the mismatch between training and inference, previous works attempt to directly optimize the task-specific metric at test time. Ranzato et al. (2015) propose sequence-level training algorithm in reinforcement learning. The models receive rewards until the completion of the entire sequence. Considering that the search space in sequence generation is exponentially large, *i.e.*, $O(|\mathcal{V}|^T)$, where $\mathcal{V}$ is a set of tokens in the vocabulary ($\sim$ 10K or more) and

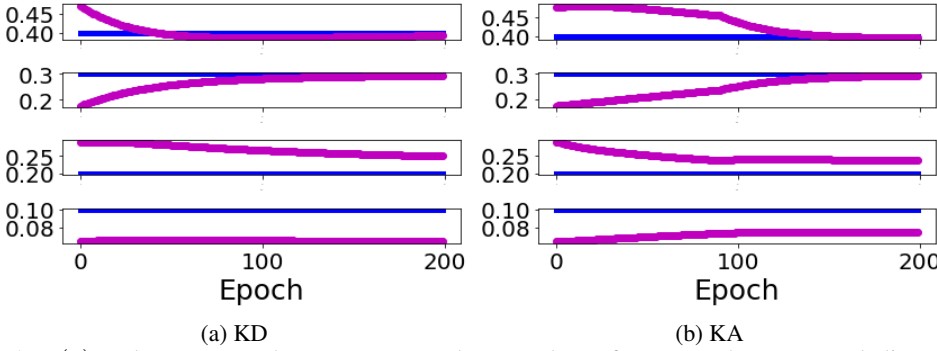

(a) KD                 (b) KA

Figure 3: $\mathbf{p}(x)$ evolves over epochs. $\mathbf{p}_0$, $\mathbf{p}_1$, $\mathbf{p}_2$ and $\mathbf{p}_3$ are shown from top to bottom. Purple lines denote $\mathbf{p}(x)$ and blue lines denote ground-truth $\mathbf{q}(x)$. $\mathbf{p}_0$ and $\mathbf{p}_1$ in KD converge to $\mathbf{q}_0$ and $\mathbf{q}_1$ in $\sim 30$ epochs, while $\mathbf{p}_2$ and $\mathbf{p}_3$ stay far away from the ground-truth $\mathbf{q}_2$ and $\mathbf{q}_3$ with respect to those in KA.

Table 1: Model configurations for WMT'17 De-En and IWSLT'15 Th-En

|  |  | encoder | | | decoder | | |
| --- | --- | --- | --- | --- | --- | --- | --- |
|  |  | layer | dim | head | layer | dim | head |
| De-En | student | 6 | 512 | 4 | 1 | 512 | 4 |
|  | teacher | 6 | 1024 | 16 | 6 | 1024 | 16 |
| Th-En | student | 1 | 128 | 1 | 1 | 128 | 1 |
|  | teacher | 3 | 256 | 2 | 1 | 256 | 2 |

$T$ is the length of the sequence ($\sim 20$ or more), the rewards are extremely sparse. This makes the training unstable. To alleviate the sparse rewards problem, Liu et al. (2017) use Monte Carlo rollouts to estimate immediate rewards at each position. Unfortunately, the estimation is very noisy and of high variance due to the exponentially large search space. Moreover, the training is computationally expensive. In this paper, we attempt to use Actor-Critic algorithms, where the "Critic" is a teacher model used to estimate action-value function, *i.e.*, the rewards of producing the current token given previous tokens, and the "Actor" is a student model that learns to produce a token at each position.

To deal with exposure bias, Bengio et al. (2015) propose to gradually replace tokens from human references to generated tokens during training. The training starts with tokens from human references and ends up with using generated tokens. However, the rewards which are the matching n-grams with a few human references limits the capability of the models in exploration. In this paper, we equip the models with knowledgeable teachers which offer smart advice based on semantic meanings.

**Transfer learning.** In the context of sequence generation, pre-training with monolingual data (Radford et al., a;b) has had significant success in language understanding. Another line of work is knowledge distillation (KD) (Hinton et al., 2015), in particular when models are of different architectures and sizes. Kim & Rush (2016) propose two levels of knowledge distillation. At sequence-level, student models are trained on the augmented dataset with outputs of running beam search with teacher models. At token-level, they get the conditional probability of each token given preceding tokens closer to that of teacher models. In other words, this is equivalent to minimizing $\mathbb{D}_{KL}(\mathbf{q}\|\mathbf{p})$, where $\mathbf{q}$ is the distribution of teacher models and $\mathbf{p}$ is the distribution of student models. Yu et al. (2017) use adversarial training to encourage the models producing human-like sequences by learning a sequence-level discriminator to distinguish generated sequence and human references. In fact, this is equivalent to minimizing Jensen-Shannon Divergence (JSD, *i.e.*, $0.5*\mathbb{D}_{KL}(\mathbf{q}\|\mathbf{p})+0.5*\mathbb{D}_{KL}(\mathbf{p}\|\mathbf{q})$), but at sequence-level. In this paper, we offer a thorough analysis of the effects of minimizing $\mathbb{D}_{KL}(\mathbf{p}\|\mathbf{q})$ on avoiding exposure bias, and we find that minimizing $\mathbb{D}_{KL}(\mathbf{p}\|\mathbf{q})$ performs the best among $\mathbb{D}_{KL}(\mathbf{q}\|\mathbf{p})$ and JSD.

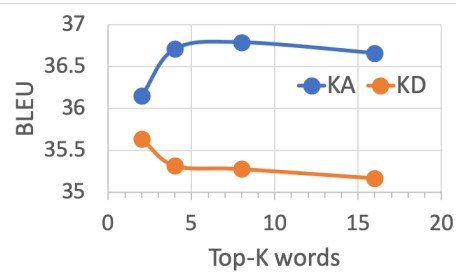 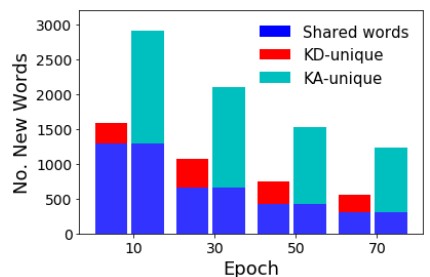

(a) Accuracy vs. Top-$k$        (b) No. of novel tokens

Figure 4: Ablation study on WMT'17 De-En task. (a) Accuracy on validation set with variable $k$. Larger $k$ means a larger part of distribution is observed. (b) Number of novel tokens emerge in the top-16. High number indicates strong exploration in search space. We noticed that ~80% probability mass are put on the top-16 tokens.

Table 2: BLEU scores on WMT'17 De-En and IWSLT'15 Th-En. (On test set)

|       | Student | Teacher | KD    | **KA**  | 1/2KA+1/2KD |
|-------|---------|---------|-------|---------|-------------|
| De-En | 30.23   | 34.10   | 31.13 | **32.24** | 31.51       |
| Th-En | 12.37   | 17.55   | 12.70 | **13.36** | 12.93       |

## 5 EXPERIMENTS

### 5.1 DATASET

We test our approach on WMT 2017 German-English with 4M sentence pairs, validate on new-stest2016 and test on newstest2017. All the sentences are first tokenized with moses tokenizer and then segmented into 40K joint source and target byte pair encoding (Sennrich et al., 2015). Another Thai-English dataset comes from IWSLT 2015. There are 90k sentence pairs in training and we take 2010/2011 as the dev set and 2012/2013 as the test set. Byte pair encoding is of size 10K.

### 5.2 TRAINING

Without a good starting point, the performance of minimizing $\mathbb{D}_{KL}(\mathbf{p}\|\mathbf{q})$ degrades because the models are more than likely stuck on the current best tokens and unlikely to explore. We therefore pre-train student models and fine-tune them in all the experiments by minimizing

$$\mathcal{L}_{ALL}(\mathbf{y};\mathbf{x},\theta,\phi) = (1-\lambda)\mathcal{L}_{NLL}(\mathbf{y};\mathbf{x},\theta) + \lambda\sum_t \underbrace{\mathbb{D}_{KL}\Big(\mathbf{p}_\theta(\mathbf{y}_t\mid\mathbf{h}_t)\|\mathbf{q}_\phi(\mathbf{y}_t\mid\mathbf{h}_t)\Big)}_{\mathbb{D}_{KL}(\mathbf{q}_\phi\|\mathbf{p}_\theta)\text{for KD}} \qquad (12)$$

where, $\lambda$ is trade-off parameters. Basically, "Critic" and "Actor" are optimized simultaneously. However, in this paper, we simply freeze teacher models and leave joint training to future work.

**Model.** Our teacher models and students models all use the transformer architecture, which has achieved state-of-the-art accuracy and is widely used in recent NMT research. Model configurations are listed in Table.1. We train all transformer models using the implementation in `Fairseq` (Ott et al., 2019). We use Adam optimizer (Kingma & Ba, 2014) with $\beta_0 = 0.9$, $\beta_1 = 0.98$, $\epsilon = 1e^{-8}$. Learning rate is 0.0001 and dropout rate is 0.3. At inference time, we use beam search with a beam size of 6 and length penalty 1.0.

### 5.3 RESULTS

Our results on NMT tasks are reported in Table.2. We tune hyper-parameters $\lambda$ and top-$k$ on validation set where $\lambda = 0.5$ for KD which is consistent with that in Kim & Rush (2016). We observed that KA consistently outperforms KD on both De-En (high-resource) and Th-En (low-resource) tasks by 0.7 - 1.1 BLEU score. In addition, we also test JSD (*i.e.*, $\frac{1}{2}$ KA + $\frac{1}{2}$ KD), which is equivalent to GAN. The accuracy lies between KA and KD. The results say that KA does help avoiding exploration bias and further close the gap between training and inference.

**Exploration.** Similar to Sec.3.2, we evaluate the performance by allowing only top-$k$ tokens to be available. We vary $k$ from 2 to 16 and conduct the experiments on validation set. In Fig. 4a, we

Table 3: Top-10 tokens in KD and KA over epochs. ([A] denotes the token the models are trying to produce at the current position and **B** is the unique token included in one approach but not the other)

| | SRC: Ich werde mich in dieser Woche darum kmmern. TRG: I will [attend] to it this week | SRC:Er knnte sich nicht frei bewegen. TRG: He could [not] move freely. |
|---|---|---|
| KD | look, be, take, deal, do, work, care, consider, address, make look, be, take, deal, work, do, make, consider, care, get look, be, take, deal, work, do, care, make, consider, **concern** ok, be, take, deal, make, work, do, care, get,**concern** look, be, take, deal, work, do, make, care, get, **concern** | not, be, do, move, 't, never, hardly, avoid, have, remain not, be, do, move, no, avoid, never, **stop**, have, also not, be, avoid, do, move, no, never, **stop**, make, have not, be, move, do, never, no, avoid, have, make, **stop** not, be, move, do, avoid, no, make, never, have, also |
| KA | look, be, take, deal, do, work, care, consider, address, make look, take, be, deal, care, work, do, make, **try**, consider take, look, be, deal, care, work, make, consider, do, **give** look, be, take, deal, work, make, consider, do, care, **try** look, be, take, deal, make, work, **see**, do, get, consider | not, be, do, move, 't, never, hardly, avoid, have, remain not, be, move, avoid, never, do, no, **fail**, have, **resist** not, move, be, never, hardly, avoid, do, make, no, also not, move, be, avoid, never, hardly, no, do, **refrain**, **fail** not, move, be, avoid, never, no, hardly, do, make, **go** |

see that the distribution is noisy because the accuracy of either KD or KA eventually goes down and when using the full information, *i.e.*, $k = |\mathcal{V}|$, KD achieves 35.41 BLEU while KA achieves 35.79 BLEU, which are far away from the best. Moreover, KA is able to learn more from noise because KD directly goes down as $k$ increases, while the accuracy of KA goes up first and then drops after $k = 8$.

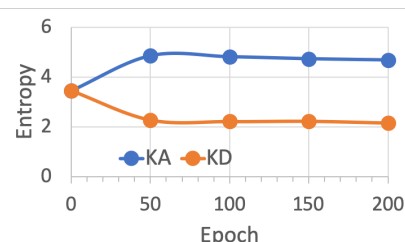

Figure 5: Entropy over epochs.

To further analyze the capability in exploration, we attempt to count the tokens in the top-$k$ which have never been included in previous epochs.

$$\left| \{ \mathbf{y}_t \in \text{Top-}k \ \mathbf{p}_\theta^i(\mathbf{y}_t|\mathbf{h}_t) \} \right.$$
$$\left. \& \{ \mathbf{y}_t \notin \text{Top-}k \ \mathbf{p}_\theta^j(\mathbf{y}_t|\mathbf{h}_t), \forall j < i \} \right|$$

where the superscript $i$ denotes epoch. We randomly sample 2000 sentence pairs from WMT'17 De-En training data. In Fig. 4b, we see that there are much more novel tokens in KA than that in KD. Table.3 demonstrates two examples. a) Given the prefix tokens "I will" and the source sentence, KA probably imagines

`try` to attend? `give` attention? `see` to?

which provide different ways to express the meaning of `attend`, while KD just try to explore `concern`. b) when predicting the token `no`, KA proposes a set of negative tokens

`fail, resist, refrain`

, and even `go` which may relate to the next token `move`.

Fig. 5 shows that the entropy of KA goes up while the entropy of KD dropping during fine-tuning. This also indicates that KA motivates exploration more than KD.

## 6 CONCLUSION

We proposed a learning principle where student models actively ask for advice from teacher models. The theoretical analysis proves that it helps to alleviate exposure bias in sequence generation and improve the capability of generalization on unseen data. In general, our approach is extendable to phase-level or sequence-level, but we simply focus on token-level due to practical issues. However, our token-level KA approach is complementary to sequence-level KD by using the outputs of teacher models as extra data. Experimental results on benchmark translation data sets show that our approach improves over previous methods of leveraging large models (or ensembles) to improve the quality of smaller, more efficient models.

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

# A    APPENDIX

## A.1    DERIVATIVES

- $\mathbb{D}_{KL}(\mathbf{q}(x)\|\mathbf{p}(x))$. The objective is

$$\text{Minimize} \sum \mathbf{q}(x) \log(\frac{\mathbf{q}(x)}{\mathbf{p}(x)}), \ s.t. \sum \mathbf{p}(x) = 1 \tag{13}$$

We write the Lagrangian for Eqn.13 as

$$L_{\mathbf{q}\|\mathbf{p}}(x, \lambda) = \sum \mathbf{q}(x) \log(\frac{\mathbf{q}(x)}{\mathbf{p}(x)}) + \lambda\left(\sum \mathbf{p}(x) - 1\right) \tag{14}$$

Method of Lagrangian multipliers involves setting the derivative of $L_{\mathbf{q}\|\mathbf{p}}$ w.r.t $\mathbf{p}(x)$ to 0,

$$G_{\mathbf{q}\|\mathbf{p}} = \frac{\partial L_{\mathbf{q}\|\mathbf{p}}}{\partial \mathbf{p}(x)} = \lambda - \frac{\mathbf{q}(x)}{\mathbf{p}(x)} = 0 \tag{15}$$

Using the fact that $\sum \mathbf{p}(x) = 1$, we can show that $\lambda = 1$.

- $\mathbb{D}_{KL}(\mathbf{p}(x)\|\mathbf{q}(x))$. The objective is

$$\text{Minimize} \sum \mathbf{p}(x) \log(\frac{\mathbf{p}(x)}{\mathbf{q}(x)}), \ \ s.t. \sum \mathbf{p}(x) = 1 \tag{16}$$

We write the Lagrangian for Eqn.16 as

$$L_{\mathbf{p}\|\mathbf{q}}(x, \lambda) = \sum \mathbf{p}(x) \log(\frac{\mathbf{p}(x)}{\mathbf{q}(x)}) + \lambda\left(\sum \mathbf{p}(x) - 1\right) \tag{17}$$

Method of Lagrangian multipliers involves setting the derivative of $L_{\mathbf{p}\|\mathbf{q}}$ w.r.t $\mathbf{p}(x)$ to 0,

$$G_{\mathbf{p}\|\mathbf{q}} = \frac{\partial L_{\mathbf{p}\|\mathbf{q}}}{\partial \mathbf{p}(x)} = 1 + \lambda + \log\frac{\mathbf{p}(x)}{\mathbf{q}(x)} = 0 \tag{18}$$

Using the fact that $\sum \mathbf{p}(x) = 1$, we can show that $\lambda = -1$.

## A.2 PROPERTY

We'll prove the key properties in Sec.3.1.2.

(a) When $\mathbf{q}(x) > \mathbf{p}(x)$, we have $1 - \frac{\mathbf{q}(x)}{\mathbf{p}(x)} < 0$ and $\log\frac{\mathbf{p}(x)}{\mathbf{q}(x)} < 0$. And, when $\mathbf{q}(x) < \mathbf{p}(x)$, we have $1 - \frac{\mathbf{q}(x)}{\mathbf{p}(x)} > 0$ and $\log\frac{\mathbf{p}(x)}{\mathbf{q}(x)} > 0$.

(b) Let's first consider the function $z - \log z$

$$z - \log z \begin{cases} > 1 & \text{if } z \neq 1 \\ = 1 & \text{if } z = 1 \end{cases} \tag{19}$$

It's easy to prove because when $z < 1$, the gradient $1 - \frac{1}{z} < 0$ and when $z > 1$, the gradient $1 - \frac{1}{z} > 0$. Thus, the function reaches the minimum value 1 at $z = 1$.

When $\mathbf{q}(x) > \mathbf{p}(x)$, we have

$$\left|G_{\mathbf{q}\|\mathbf{p}}(x)\right| - \left|G_{\mathbf{p}\|\mathbf{q}}(x)\right| = \left|1 - \frac{\mathbf{q}(x)}{\mathbf{p}(x)}\right| - \left|\log\frac{\mathbf{p}(x)}{\mathbf{q}(x)}\right| \tag{20}$$

$$= \frac{\mathbf{q}(x)}{\mathbf{p}(x)} - \log\frac{\mathbf{q}(x)}{\mathbf{p}(x)} - 1 > 0 \tag{21}$$

When $\mathbf{q}(x) < \mathbf{p}(x)$, we have

$$\left|G_{\mathbf{q}\|\mathbf{p}}(x)\right| - \left|G_{\mathbf{p}\|\mathbf{q}}(x)\right| = \left|1 - \frac{\mathbf{q}(x)}{\mathbf{p}(x)}\right| - \left|\log\frac{\mathbf{p}(x)}{\mathbf{q}(x)}\right| \tag{22}$$

$$= 1 - \left(\frac{\mathbf{q}(x)}{\mathbf{p}(x)} - \log\frac{\mathbf{q}(x)}{\mathbf{p}(x)}\right) < 0 \tag{23}$$

