# OpenReview forum: "Proactive Sequence Generator via Knowledge Acquisition"
_ICLR.cc/2020/Conference — Reject_

### Official Review · AnonReviewer3 · 2019-10-23
**Official Blind Review #3**

**Rating:** 3

**Review:**

This paper addresses the problem of training small models to mimic large models but, in constrast to knowledge distillation, minimize the reverse-KL between the teacher and the model instead of the forward-KL.

The authors notice that there is an interesting interaction between beam search (i.e. focusing only on the top-k tokens) and distillation. By minimizing the forward-KL, distillation focuses the student on having the non-negative mass on all the words selected by the teacher. However, the authors argue that minimizing the reverse-KL makes more sense: to only include tokens in the students that are present in the teacher.

The paper then spends time explaining the difference between minimizing the KL (KD) or the reverse KL (their proposed KA) and show some experiments validating their methods.

I think the paper is well-written, but can be sometimes difficult to follow. For example, they introduce the notation p_\theta and q_\phi without specifying who is the teacher and who is the student (I assumed that q was the teacher and p the student as in the introduction). The paper spend a bit of time explaining the qualitative difference between minimizing the KL or the reverse-KL. Even though it is useful, I believe it is well known in the community and can be found in multiple standard references (e.g. (Murphy, 2012) or this online class on graphical models: https://ermongroup.github.io/cs228-notes/inference/variational/). I don't think this constitutes a contribution yet the authors spend a fair amount of the paper on that particular topic.

The idea is quite simple but seems to be effective (up to +1.9 BLEU on German to English and +0.6 BLEU on Thai to English). I think it would have been useful to say how each model were tuned for fair comparison (e.g. how was the learning rate chosen?). I would have also like to see more tasks, like language modelling, question answering or text summarization.

I also think the use of the term `actor-critic' is misleading given that, as far as I understand, there is no reinforcement learning in this paper. Section 3.1.2 is really confusing: are you referring to the derivative of the KL between two finite-dimensional vectors? Is there a lagrangian because you are somewhat taking the derivative on the simplex?

Overall, this paper proposes an interesting trick that seems to work in practice but the novelty remains limited.

(Murphy, 2012) Machine Learning: a Probabilistic Perspective. Murphy, Kevin. 2012.

**Experience Assessment:**

I have published one or two papers in this area.

**Review Assessment: Checking Correctness Of Derivations And Theory:**

I assessed the sensibility of the derivations and theory.

**Review Assessment: Checking Correctness Of Experiments:**

I assessed the sensibility of the experiments.

**Review Assessment: Thoroughness In Paper Reading:**

I read the paper at least twice and used my best judgement in assessing the paper.

---

> ### Comment · AnonReviewer3 · 2019-11-15
> **--**
>
> I have read the authors' rebuttal and they only limitedly addressed my concerns. I will maintain my assessment of the paper!

---

> > ### Author Response · Authors · 2019-11-15
> > **What concerns are not addressed?**
> >
> > Thanks for reviewer's response. Could you please elaborate what concerns are not addressed? We'll try to explain more. Meanwhile, we are running experiments on language modeling task. Will try to update what we have observed before the deadline (tomorrow).

---

### Official Review · AnonReviewer2 · 2019-10-23
**Official Blind Review #2**

**Rating:** 3

**Review:**


This paper introduces Knowledge Acquisition (KA), i.e., KL-divergence in the reverse order as the loss function to train student models for sequence-to-sequence tasks, and experiments were done on WMT’17 De-En and IWSLT’15 Th-En translation tasks.

Pros:
The paper is clearly written. The authors clearly show the reason to use KL-divergence in the reverse order. They provide a concrete analysis of the effects of minimizing the proposed KA loss function to alleviating exposure bias.
Some figures like Fig. 1 and Fig. 2 helps to show the paper.

Cons:
 As the analysis of the authors, KL-divergence in the reverse order is an alternative of KL-divergence for training student models on sequence generation tasks. However, I have a little concern that the contribution is relatively limited since it is known that KL-divergence is not symmetric and the proposed KA (KL-divergence in the reverse order) may be thought a little straightforward. In addition, KA is only investigated on token-level, as the authors said, “due to practical issues”.
The experiments in this paper may be thought insufficient. Specifically, the authors only establish teacher and student model baselines by themselves. It is reasonable that the proposed method could be compared with other similar KD methods, like [1] mentioned in the paper. Another concern is whether the proposed KA method can be applied in current sequence-to-sequence state-of-the-art teacher models, such as GPT-2 [2] and XLM [3], and whether it is still effective.


Reference
[1] Yoon Kim, Alexander M. Rush. Sequence-Level Knowledge Distillation. EMNLP 2016
[2] Alec, Radford, Jeffrey Wu, Rewon Child, David Luan, Dario Amodei, and Ilya Sutskever. Language models are unsupervised multitask learners. OpenAI Blog 2019.
[3] Guillaume Lample, Alexis Conneau. Cross-lingual Language Model Pretraining. CoRR abs/1901.07291


**Experience Assessment:**

I have published one or two papers in this area.

**Review Assessment: Checking Correctness Of Derivations And Theory:**

I assessed the sensibility of the derivations and theory.

**Review Assessment: Checking Correctness Of Experiments:**

I assessed the sensibility of the experiments.

**Review Assessment: Thoroughness In Paper Reading:**

I read the paper at least twice and used my best judgement in assessing the paper.

---

### Official Review · AnonReviewer1 · 2019-10-25
**Official Blind Review #1**

**Rating:** 3

**Review:**

[Overview]

In this paper, the authors proposed a new method called knowledge acquisition (KA) for distilling the learned knowledge from the teacher model to the student model. Unlike the conventional KL-divergence based knowledge distillation method, the authors take the reverse version which learns the student to increase the precision. The paper gave a thorgough analysis on the proposed KA strategy and compared with other strategies like KL and JS divergence. On the sequence generation task (translation), the authors showed that the proposed KA strategy achieved better performance compared with KD based methods when distilling the knowledge from a teacher model to a student model.

[Pros]

1. The authors proposed a new strategy to perform the knowledge distillation from a teacher model to student model for sequence generator. To improve the precision of the student model, the authors proposed to invert the formula of KL divergence, i.e., the position of prediction probability from teacher and student models.

2. The authors presented a thorough analysis on the proposed KA strategy and compared it with KL strategy in terms of the precision and recall for the student models. I think it is very readable and understandable. This analysis align with those put on generative adversarial network.

3. The authors performed the experiments on the translation tasks showing that the proposed KA strategy outperforms both KL and JS strategy in terms of the generation performance. Also, the authors ablated the number of top reference tokens from the teacher model and showed that using a reasonable number of top tokens is important to help alleviate the noised in the teacher model.

[Cons]

The main concern about the proposed method is whether it can be used as a generic strategy for transferring the knowledge from the teacher model to student model.

1. First, I have a doubt on the stability of the proposed strategy. In my opinion, the improvements on the sequence generation tasks are mainly due to the tuned hyper parameters for the training, especially the lambda in Eq(12), which is tuned at the validation set. It controls how much to modulate the prediction distribution of student model toward that of teacher model. However, a less tuned lambda would cause either over curve fitting or under curve fitting. As a result, the authors should: 1) first show how the performance would be affected by varying the lambda in the formula; 2) from the reading, I did not see whether the lambda was tuned as well for KD or (KD + KA) / 2. If not, then for fair comparison, the authors should tune the lambda for the KD and (KD + KA) / 2 strategy as well. At some point, I would think the combination of KD and KA would be better than either of them.

2. Second, some experimental results are somewhat counter-intuitive to me. These are two folds: a) In Figure 4(b), we can see that the KA strategy has learned to generate more new tokens compared with KD. This is a bit strange to me because, KA will focus on the precision instead of recall. To me, pushing the precision will high likely sacrifice the recall and thus the number of novel tokens generated by the model. b) similarly, in Figure 5, it is shown that KA has generally higher entropy than KD. This is also a bit counter intuitive. In Eq(6), it is obvious that the proposed strategy has a entropy term which will be reduced when we want to reduce the KL divergence during the training time. From Figure 5, KA and KD start from the same point (I guess it is because the same pre-trained student model) is used. However, for KA, the entropy start to increase and then converge to a stable number which is consistently higher than KD strategy.

3. Third, Figure 4(a) also indicates some thing. When only the top few tokens are used to transfer the knowledge from teacher model to student model, KA focus on the precision of a small subspace, which tends to have few modes. However, when the number of tokens is increased, the mode number would also increase drastically. i guess that’s why the both strategies finally become very close to each other, and the minor gap between them is probably due to the benefit from hyper-parameter fine-tuning.

Besides the above comments. there are some minor points which are missed in the paper:

1. As pointed above, it is not clear whether the same tuning is also applied to KD and (KD + KA) / 2. the authors should mention this in the experiment section.

2. It is also not clear how many tokens is used for reporting the numbers in Table 2. Is it the whole vocabulary side? If this is the case, the gap between KA and KD on validation set are pretty close while more significant on test set.

3. In Eq (11), should there be a minus sign before the expectation?

4. Also, is there any more comment on why it is hard to train the student model joint from scratch? what will happen in this case?

[Summary]

In this paper, the authors proposed a new strategy called Knowledge Acquisition which is used for distilling the knowledge learned from  teacher model to the student model. Different from KD strategy, it inverted the position of probability distributions for teacher and student models. By this way, the KA strategy learns a student model which can achieves higher precision. The proposed strategy is evaluated on sequence generation, particularly translation task.  However, as pointed above, in my opinion, there are some counter-intutive observations in the experimental results. It would be good if the authors can address these concerns in the rebuttal.

**Experience Assessment:**

I have read many papers in this area.

**Review Assessment: Checking Correctness Of Derivations And Theory:**

I assessed the sensibility of the derivations and theory.

**Review Assessment: Checking Correctness Of Experiments:**

I assessed the sensibility of the experiments.

**Review Assessment: Thoroughness In Paper Reading:**

I read the paper thoroughly.

---

### Author Response · Authors · 2019-11-06
**The minus sign in Eq.6**

We would like to thank the reviewers for their thoughtful comments and efforts. Before addressing concerns of reviewers, we'd like to clarify a typo in Eq.6.

The minus sign in Eq.6 is for the entire expression. That means, it should be - H[p(student)]. Minimizing KL(student | teacher) is equivalent to maximizing expected rewards + entropy. Thus, KA encourages exploration which is consistent with what we observed in experiments.

Hope this is able to address the main concern of R1.

---

### Author Response · Authors · 2019-11-06
**Contribution**

We thank the reviewers for their thoughtful comments and efforts again! To address a general concern of reviewers, we'd like to elaborate our contribution:

There is a thorough analysis on KL-divergence in the literature [e.g., Sec 21.2.2, page 733, in Murphy’s book]. The basic idea is:
[1] Assume p (real distribution) is multimodal and q (surrogate distribution) is unimodal
[2] KL(i.e., KL(p||q)) is zero-avoiding for q and the resulting modes of q will be in low density, right between modes of p. [q “covers” p]
     reverse-KL (i.e., KL(q|p)) is zero-forcing for q, and q locks on a single mode

The insight has been widely used in a variety of research areas such as variational inference and GAN. However, the difference in learning strategies is still unclear.  Moreover, unimodal distribution constraint in [1] doesn’t hold anymore when q is parameterized by a neural network.

Instead, we investigate the derivatives of their objectives and then compare learning strategies given gradient-based optimizers. The key takeaway is
    KL: optimize recall
    reverse-KL: encourage exploration; optimize precision

The conclusion can be generalized to any tasks involving KL-divergence in training deep neural networks (typically using gradient-based optimizers).

We test our hypothesis on NMT tasks.  NMT is representative of sequence generation tasks including language modeling and text summarization. The difference is just the type of inputs and outputs. We observe that KA outperforms KD and our gradient-based analysis explains that this is because reverse-KL helps to mitigate exposure bias which is known to be a key challenge in training sequence models.

In summary, we are the first to analyze the behavior of KL and reverse-KL based on gradients which is important to unveil the learning strategy in training deep neural network. Moreover, our analysis doesn't enforce any constraints on p and q. Thus, the conclusion is generalizable.

---

### Author Response · Authors · 2019-11-06
**address technical concerns**

We thank the reviewers for their thoughtful comments and efforts again! We'll address all technical concerns with more details.

# Reviewer 1

1)  \lambda is tuned for KD and (KD+KA)?

Yes, \lambda is tuned on validation set. For KD, we observed \lambda=0.5 which is consistent with the original KD paper.

2) KD+KA would be better than either of them?

It’s intuitive that KD+KA performs better because of the trade-off. However, we do observe that KA performs the best. This indicates that exploration does matter a lot in order to mitigate the distribution mismatch between learning and inference.

3) Why KA learns more new tokens?

The entropy term in Eq.6 (after correction) explains this. At test time, when feeding generated tokens (which probably far away from ground truth), the student models need to make decision in under-explored search space. Thus, exploration over new tokens during training helps to reduce accumulated errors (called exposure bias)

4) KA has higher entropy?

Maximize Eq.6 -> maximize entropy

5) KA and KD converge together?

Ideally, KA and KD converge to the same equilibrium state where p=q if the student distribution is parameterized by a neural network with infinite capacity. However, in practice, teacher distribution is noisy, and with limited capacity, the optimal solution is unreachable. Empirically, we observe that KL (or reverse-KL) term is not zero when the training converges. In general, different learning strategies do matter!


6) It is also not clear how many tokens is used for reporting the numbers in Table 2. Is it the whole vocabulary side?

We tune top-k on validation set as well. Will update the paper with more details.

7) In Eq (11), should there be a minus sign before the expectation?

We try to maximize Eq.11 which is an extension of Eq.6.

8) Also, is there any more comment on why it is hard to train the student model joint from scratch? what will happen in this case?

We are able to train teacher models and student models together like what standard actor-critic algorithm does. To be simple, we don’t explore in the current paper since we mainly focus on the analysis that why reverse-KL helps to avoid exposure bias.

# Reviewer 2

1)  Use state of the art models?

The teacher models (Big Transformer with L=6, D=1024, H=16) on WMT de-en task are already the state-of-the-art. For low-resource language pair th-en, we use small Transformer models to avoid over-fitting.

Both GPT-2 and XLM are language models. Their goal is to learn good language representation and basically used as pre-training. And, both of them use Transformer architecture as well.


# Reviewer 3

1)  `actor-critic' is misleading

We prove that minimizing KL(p(student)||q(teacher)) is equivalent to maximizing actor-critic + entropy , which helps to explain why KA motivates sequence generator to be proactive and explore more. Moreover, the equivalence opens a door to leverage techniques from two domains to further improve the model.

2) referring to the derivative of the KL between two finite-dimensional vectors?

In sequence generation tasks, p and q are distributions over tokens in the vocabulary (e.g., 10k)

3)  Is there a lagrangian because you are somewhat taking the derivative on the simplex?

Yes, please refer to Appendix A.1 for the proof.

Will update the paper with more details and more experiments!

---

### Author Response · Authors · 2019-11-15
**Experiments on language modeling tasks**

The difference in objectives shows that KL(student | teacher) encourages the exploration a lot, which results in much smooth distributions that are able to capture multiple modes. That means, the probabilities of words with similar semantic meanings should be pushed up. Thus, the probability of ground truth word (representative of a single mode) should be lower.

We do observe such thing on language modeling tasks (WikiText-103). We found that KL(student | teacher) gives higher perplexity compared with KL(teacher | student).
           Setting: top-16, \lambda=0.3
                 Student: Big transformer -> 34.76
                 Teacher: GPT2-small -> 23.36
                 KL(student | teacher) -> 33.40
                 KL(teacher | student) -> 32.76

Due to the limited time, we don't get enough time to sweep hyper-parameters exhaustively on validation set, but a bunch of experiments give the same signal.

This is consistent with what we observed on NMT tasks, where when training converges, KL(teacher | student) gives 5.38 ppl and KL(student | teacher) gives 6.32 ppl. However, the latter gives much higher bleu score on test set.  We believe that the smooth distributions produced by KL(student | teacher) would help to generate diverse sentences (not overfitting to human references) and enable the language models adapt to downstream tasks quickly and easily.

Paper [1] published on ICML'18 also claim that capturing multi-modal distribution is important in sequence generation tasks. (please check out references for more relevant papers)

[1] Myle Ott, Michael Auli, David Grangier, Marc'Aurelio Ranzato. Analyzing Uncertainty in Neural Machine Translation, ICML 2018

---

### Decision · Program_Chairs · 2019-12-19

**Decision:**

Reject

**Comment:**

This paper shows a nice idea to transfer knowledge from larger sequence models to small models. However, all the reivewers find that the contribution is too limited and the experiments are insufficient. All the reviewers agree to reject.